# The Effect of MUFA-Rich Food on Lipid Profile: A Meta-Analysis of Randomized and Controlled-Feeding Trials

**DOI:** 10.3390/foods11131982

**Published:** 2022-07-05

**Authors:** Xinyi Cao, Jiayue Xia, Yuhao Zhou, Yuanyuan Wang, Hui Xia, Shaokang Wang, Wang Liao, Guiju Sun

**Affiliations:** 1Key Laboratory of Environmental Medicine and Engineering of Ministry of Education, School of Public Health, Southeast University, Nanjing 210009, China; 220214004@seu.edu.cn (X.C.); 220203869@seu.edu.cn (J.X.); 220213987@seu.edu.cn (Y.Z.); wyypro@foxmail.com (Y.W.); huixia@seu.edu.cn (H.X.); shaokangwang@seu.edu.cn (S.W.); gjsun@seu.edu.cn (G.S.); 2Department of Nutrition and Food Hygiene, School of Public Health, Southeast University, Nanjing 210009, China; 3China-DRIs Expert Committee on Macronutrients, Beijing 100052, China

**Keywords:** mono-unsaturated fatty acids, lipid profile, randomized controlled trial, meta-analysis

## Abstract

Since the effects of mono-unsaturated fatty acids (MUFA) on lipid profile are still controversial, a meta-analysis of randomized controlled trials was conducted in the present study to assess the effect of MUFA-rich food on lipid profiles. The study was designed, conducted, and reported according to the guidelines of the 2020 preferred reporting items for systematic reviews and meta-analysis (PRISMA) statement. A systematic and comprehensive search was performed in several databases from inception up to 30 January 2022. The results showed that the intake of edible oil-derived MUFA (EDM) could increase the blood HDL-C level (mean difference = 0.08; 95%CI: −0.01, 0.17, *p* = 0.03), but did not affect the level of TC, TG, or LDL-C. Moreover, the consumption of other food-derived MUFA (ODM) significantly decreased TG concentration (mean difference = −0.35; 95%CI: −0.61, −0.09, *p* = 0.01)), but did not affect the level of TC, LDL-C, or HDL-C. Findings from this study suggest that MUFA-rich food might be beneficial to modulate the blood lipid profile.

## 1. Introduction

Dyslipidemia, clinically featured by abnormally high blood lipid concentrations, is frequently associated with atherosclerotic heart disease, including coronary heart disease, stroke, and peripheral artery disease [1,2,3]. Evidence from 1127 population-based studies measuring blood lipids in 102.6 million people aged 18 years and older showed that although there was little change in total cholesterol, or non-HDL lipoprotein cholesterol in the global population from 1980 to 2018, the net effect in low- and middle-income countries, particularly in East and Southeast Asia, increased but declined in high-income western countries, particularly in Northwest Europe, and in Central and Eastern Europe. In 2017, non-HDL lipoprotein cholesterol caused approximately 3.9 million deaths worldwide, half of which occurred in East, Southeast, and South Asia [4]. Data from a survey in 2014 on blood lipids in the Chinese population showed that the prevalence of dyslipidemia was similar in rural and urban Chinese participants, with 43.2% and 43.3%, respectively. Moreover, urban individuals had a higher prevalence of low HDL-C than the residence living in the rural areas (20.8% versus 19.2%). The prevalence of elevated LDL-C (7.8% vs. 8.3%), elevated TC (10.9% vs. 11.8%), and elevated non-HDL-C (10.0% vs. 10.9%) was lower in urban residents. Furthermore, women were more likely than men to have elevated TC, LDL, and non-HDL-C [5].

Studies have shown that dyslipidemia is associated with an increased incidence of diabetes-related complications as well as the development of cancer [6]. In addition, studies have shown that the lipid levels of 60–70% of adults are outside of the recommended ranges [7]. Since dyslipidemia is asymptomatic in its early stages, dyslipidemia often goes undiagnosed until patients develop vascular complications [8]. Therefore, dyslipidemia has become a major public health problem in the world [9].

Mono-unsaturated fatty acids (MUFA) are fatty acids with single double bond [10]. In contrast to saturated fatty acids, which are thought to raise cholesterol levels, poly-unsaturated fatty acids could be beneficial to lower cholesterol levels, while MUFA have long been thought to have a neutral effect on plasma cholesterol levels. However, epidemiological studies found that regular dietary intake of MUFA-rich food such as olive oil and avocado could reduce the risk of coronary heart disease by ameliorating dyslipidemia [11,12]. In addition, MUFA intake is particularly high, the serum cholesterol level and the incidence of coronary heart disease is and are relatively low in the Mediterranean countries, which is probably due to the high intake of MUFA in these countries [13]. Therefore, the potential beneficial effects of MUFA on blood lipids might have been ignored, and it is necessary to further explore the physiological properties of MUFA.

However, results from different research on MUFA are conflicted. A study found that adding cashews to the typical American diet reduced total cholesterol and LDL cholesterol, as compared to a control diet [14]. However, it was reported by another study that cashew nut supplementation increased HDL cholesterol levels and had no effect on TC and LDL-C concentration [15]. Moreover, the majority of previous studies on monounsaturated fatty acids were prospective cohort studies on the risk and prevalence of cardiovascular disease, as well as the effects of a MUFA-rich diet (but with interference from other components) on lipid profiles [10,12,16,17,18,19,20,21]. Therefore, it is necessary to conduct a meta-analysis to assess the effect of MUFA on lipid profiles. In this study, we integrated the various studies on the effect of MUFA-rich foods on blood lipids by collecting and synthesizing the clinical evidence followed by a meta-analysis, which aims at providing a more comprehensive view on the regulatory effects of MUFA-rich diet on blood lipids.

## 2. Materials and Methods

Our study was designed, conducted, and reported according to the guidelines of the 2020 preferred reporting items for systematic reviews and meta-analysis (PRISMA) statement.

### 2.1. Literature Search Strategy

Relevant literatures were identified via a systematic and comprehensive search from the PubMed, Web of Science, Embase, and Cochrane library databases up to 30 January 2022. We only included the literature published in English. The search used the terms combined with MeSH. The search string is as follows: (“Acid, Monounsaturated Fatty” OR “Monounsaturated Fatty Acid ” OR “Monounsaturated Fatty Acid” OR “Acid, Monounsaturated Fatty” OR “Fatty Acid, Monounsaturated ”OR “MUFA” OR “Myristoleic acid” OR “Palmitoleic acid” OR “palmitelaidic acid” OR “palmitoleate” OR “Oleic acid” OR “Erucic acid”) and (“Cholesterol” OR “HDL” OR “LDL” OR “Triglycerides” OR “Lipoprotein” OR “low-density lipoprotein” OR “LDL-C” OR “VLDL” OR “HDL-C” OR “TG” OR “TC” and (“Adults” OR “Men” OR “Women” OR “Children” OR “Adolescent”). The reference lists of each study, systematic reviews, and meta-analyses were also reviewed to detect any relevant studies that might have been missed.

### 2.2. Inclusion and Exclusion Criteria

The meta-analysis selected original studies that met the following inclusion criteria: (a) the study design must be a randomized controlled trial which investigated the effects of MUFA-rich food on blood lipids; (b) the outcome should include at least one of the changes of TC, LDL-C, HDL-C or TG level; (c) all of the data must be complete and available; (d) the dose of MUFA in the intervention group was significantly higher than that in the control group. The following exclusion criteria were used: (a) cell experiments, animal experiments, meta-analyses, reviews, conference papers, case reports, and editorials; (b) studies that clearly did not meet the above criteria.

### 2.3. Data Extraction

Two reviewers evaluated and extracted data independently from eligible articles by a specially designed data collection form. The following information was extracted: the name of the first author, year of publication, country, mean participant age, study design, the number of subjects, intervention group and control group, MUFA source, and the daily intake amount of MUFA. Moreover, standard deviation and net mean change of TC, TG, LDL-cholesterol, and HDL-cholesterol in each of the literature were also extracted.

### 2.4. Quality Assessment

The seven validity questions of the Cochrane Risk of Bias Tool were applied to evaluate the literature quality evaluation of the included studies; and the tool assessed the random sequence generation, allocation concealment, blinding of participants and personnel, blinding of outcome assessment, incomplete outcome data, selective outcome reporting, and other bias. According to the recommendations of the Cochrane Handbook, each item was judged to have a “Low”, “High”, or “Unclear” risk of bias.

### 2.5. Data Analysis

All statistical analyses were conducted using Stata 15.0, and *p* < 0.05 was considered statistically significant. The mean value of relevant indicators was calculated by subtracting the average change (final value minus baseline value). If there were more than one endpoint, the last endpoint value was used. For the parallel controlled study, we calculated the mean difference in effectiveness (MD) by comparing the change from control to baseline and the change from intervention to baseline. For the crossover test, serum lipid values at the end of the control period and at the end of the intervention period were extracted, and MD in effect was calculated. If the study did not provide a net change in SD, the net change in SD was calculated according to this formula. Since some data were not presented in a uniform way in the original literature, we converted the different units of blood lipid levels to mmol/L (1 mg/dL = 0.0258 mmol/L). In addition, several studies only provided standard error (SE) and 95% CIs of the data, so we calculated the standard deviation (SD) as appropriate. Furthermore, the mean net change was calculated by subtracting the mean value of the endpoint from the corresponding mean value at the basal line. The net change of SD was calculated using the formula: SD_change_ = (SD^2^_baseline_ + SD^2^_endpoint_ − 2R × SD_baseline_ × SD_endpoint_) 1/2, correlation coefficient R = 0.5 [22]. I^2^ metrics and chi-squared statistics were used to assess the heterogeneity. When I^2^ > 50% or *p*-value of χ^2^ test < 0.10, which indicates a high level of heterogeneity, the random-effects model was applied. Additionally, subgroup analyses were conducted based on the source of MUFA and the background information of participants. Funnel plots and Egger’s regression test were applied to investigate potential publication bias. Moreover, the sensitivity analysis was used to evaluate the influence of a single study on the overall results.

## 3. Results

### 3.1. Literature Search and Study Characteristics

The complete flow diagram of the selected studies is shown in Figure 1. In brief, 4195 articles were found with the search strategy from four databases, where 958 duplications and 3200 irrelevant articles were removed based on the title and abstract. Then, 127 trials were included to be assessed for eligibility; of those, 130 studies were excluded as they were not randomized controlled trials, or due to the absence of an appropriate control group or full access to data. Finally, seven studies were included in this meta-analysis. Among the included studies, two articles were divided into two trials, in which, two diets with low MUFA content (coca butter and soybean oil) as the comparator, and two MUFA-rich foods (avocado and oleic acid) were involved, respectively. Therefore, seven trials with ten sets of data were eligible to be included in the meta-analysis.

### 3.2. Study Characteristics

These studies were published from 2005 to 2021 and included 248 subjects. Two studies were conducted in Asia [23,24], one study was in Europe [25], one was in Africa [26], and the other five studies were from Australia [27] and America [28,29]. The mean age of the participants ranged from 11.1 to 51.42 years. The follow-up period ranged from 4 to 16 weeks. More detailed characteristics of studies were summarized in Table 1. The risks of bias of the included studies are shown in Table 2.

### 3.3. Results of Meta-Analysis

Since the MUFA involved in the included studies was derived from either edible oil or other food sources (avocado or oleic acid), all of the trials were divided into two groups including edible oil-derived MUFA (EDM) and other food-derived MUFA (ODM) for the subsequent analyses.

The results in Figure 2 show that the intake of EDM contributed to a significant increase in the HDL-C level (mean difference = 0.08; 95%CI: −0.01, 0.17, *p* = 0.03, I^2^ = 33.2%). However, there was no significant effect of EDM consumption on serum TC (mean difference = 0.01; 95%CI: −0.46, 0.48, *p* = 0.95, I^2^ = 79.4%), LDL-C (mean difference = 0.15; 95%CI: −0.02, 0.32, *p* = 0.09, I^2^ = 37.0%), or TG concentration (mean difference = −0.16; 95%CI: −0.52, 0.20, *p* = 0.40, I^2^ = 76.6%).

Different from the effects of EDM, the intake of ODM significantly decreased the serum TG level (mean difference = −0.35; 95%CI: −0.61, −0.09, *p* = 0.01, I^2^ = 6.5%). On the contrary, the effects of ODM consumption on TC (mean difference = −0.14; 95%CI: −0.33, 0.04, *p* = 0.12, I^2^ = 0%), LDL-C (mean difference = −0.10; 95%CI: −0.26, 0.05, *p* = 0.19, I^2^ = 0%), and HDL-C level (mean difference = 0.02; 95%CI: −0.04, 0.08, *p* =0.50, I^2^ = 16.9%) were insignificant (Figure 3).

### 3.4. Subgroup Analysis and Sensitivity Analysis

Since only the result of the effects of EDM showed the heterogenicity (I^2^ > 50%), we then performed the subgroup analyses with the trials in the EDM study. Reduction of the heterogeneity was observed in TC, LDL-C, and TG, which might be relevant to two factors that performed subgroup analyses (Table 3). Particularly, the source of MUFA showed a significant change in the heterogeneity between the two subgroups. However, there was a significant difference in LDL-C. Moreover, we evaluated the stability of the results by a sensitivity analysis. There was no substantial change in the results of the TC, LDL-C, HDL-C, or TG level.

### 3.5. Publication Bias

There was no publication bias in the visual inspection of the funnel plot. Meanwhile, the results of Egger’s test showed no evidence of potential publication bias in the combined results of the four effect indexes (detailed in Figure 4 and Figure 5).

## 4. Discussion

The beneficial effects of dietary unsaturated fatty acids have been recognized. While the majority of the previous studies concentrated on the protective role of PUFA in improving lipid profile, there were few studies reporting the influence of MUFA on lipid spectrum [30]. In recent years, more and more studies have explored the cardio-protective effects of dietary MUFA, but the results of these studies were inconsistent, which impelled us to conduct a meta-analysis by synthesizing the evidence from clinical trials that investigated the effect of dietary MUFA on lipid profile [31,32]. In comparison to observational studies, there are fewer subject biases in a meta-analysis because all the studies included were randomized controlled trials.

The results from the present meta-analysis showed significant increase in HDL-C following the consumption of MUFA-rich edible oil, but there was no significant change in TC or LDL-C. Such findings were consistent with previous reports which showed that the serum level of HDL-C could be increased by the consumption of high-oleic acid ground beef [33,34]. Additionally, a meta-analysis found that substituting palm oil for PUFA-rich oils significantly increased the HDL-C level [35]. However, the mechanism underlying the regulation of HDL-C by MUFA is ambiguous, which is warrant for further investigations.

Different from the effects of EDM, the effect of ODM on TG, but not HDL-C, was significant, which was consistent with previous reports showing that the serum level of TG could be reduced by MUFA-rich food in individuals with or without hypertriglyceridemia [26,36]. The underlying mechanisms of the TG-lowering effect of dietary MUFA are unclear. However, two possible mechanisms have been proposed. Firstly, MUFA could alter the composition of VLDL, which further determines the conversion of VLDL to other lipoproteins and the metabolism of triacylglycerols. Secondly, MUFA could regulate the activities and expressions of the enzymes and proteins involved in VLDL endovascular processing and catabolism, both of which can reduce serum triacylglycerol concentrations [37,38]. Interestingly, it has been reported that dietary intake of canola oil (rich in MUFA) could substantially reduce the serum levels of TC and LDL-C. In addition, palm olein was reported to have similar effects. These findings along with the results from the present study suggest the multiple roles of dietary MUFA in regulating blood lipid profile. Due to the compositions of nuts and avocados being complex, there might be other components present in the ODM group that could also beneficially affect the TG level. However, given that various studies have reported the effects of MUFA on reducing TG, it is speculated that the lowered TG level in the ODM was, at least partially, a result from the intake of MUFA [39,40,41].

Of note, there has been a lot of compelling and convincing evidence that PUFA has a TG lowering effect [42,43,44]. It was reported that the omega-3 fatty acids content of the oils could inhibit VLDL-C and apolipoprotein-B100 synthesis, which leads to the reduction of serum TG concentrations [45]. Moreover, n-3 PUFA, especially EPA and DHA, are potent natural ligands of peroxisome proliferator-activated receptor α (PPARα) and enhance fatty acid β-oxidation through a peroxisome proliferator-activated receptor (PPAR-α)-mediated pathway, thereby reducing the substrate utilization of triglyceride formation [46]. In addition, n-3 PUFA reduce triglyceride synthesis by inhibiting diacylglycerol acyltransferase, fatty acid synthase, and acetyl-CoA carboxylase [44,46]. Since we found in the present study that MUFA-rich food could reduce TG but increase HDL-C, we speculated that dietary MUFA might play similar roles as dietary PUFA in regulating the lipid spectrum. However, whether there is any overlap among the mechanisms needs to be further determined.

In the subgroup analysis, it was found that there was a significant reduction in TC, HDL-C, and TG in the subset of palm olein, while only HDL-C levels gained a significant increase in the subset rapeseed oil. This suggested that MUFA-rich food may be more effective in improving lipid profile levels in people with disease than in healthy people. The results of the subgroup analysis suggested that the source of MUFA and participants’ condition may be the sources of heterogeneity. It has to be admitted that heterogeneity between studies was high in some results and was significant in some subgroups despite our efforts to find the source of heterogeneity in the subgroup analysis. Moreover, the components of the MUFA-enriched food are so diverse that it is not possible to determine whether nutrients other than MUFA affect the lipid profile.

It has to be admitted that heterogeneity between studies was high in some results and was significant in some subgroups despite our efforts to find the source of heterogeneity in the subgroup analysis. Moreover, the components of the MUFA-enriched food are so diverse that it is not possible to determine whether nutrients other than MUFA affect the lipid profile.

## 5. Conclusions

Collectively, results from this meta-analysis indicated that the intake of edible oil- EDM could increase the level of blood HDL-C, and the consumption of ODM could significantly decrease the TG concentration. In addition, no compelling evidence of an effect of MUFA-rich food consumption on TC or LDL-C concentration was found. However, well-designed and high-quality randomized controlled trial studies with larger-scale participants are necessary to validate current findings.

## Figures and Tables

**Figure 1 foods-11-01982-f001:**
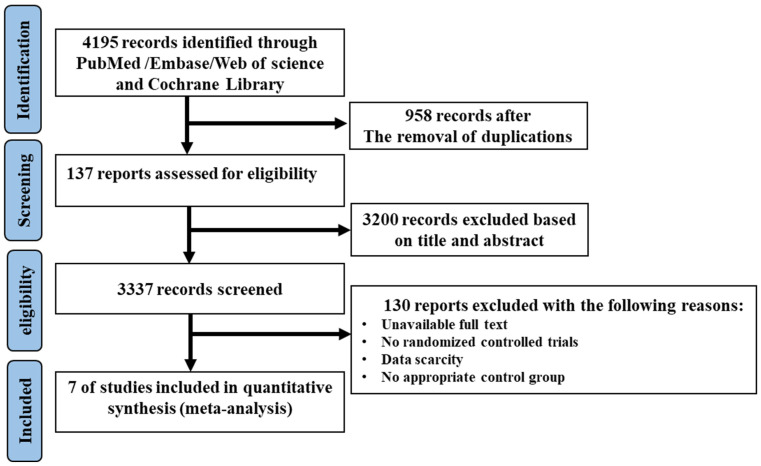
The flow diagram of trial selection.

**Figure 2 foods-11-01982-f002:**
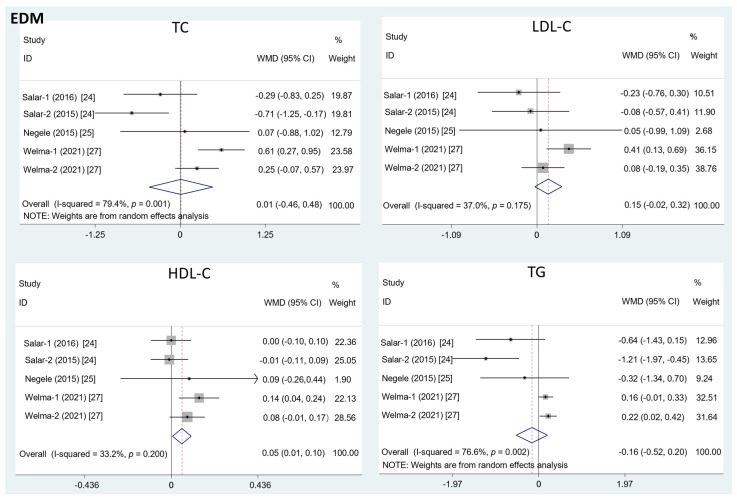
The effects of EDM on TC, LDL-C, HDL-C, and TG. TC = total cholesterol; LDL-C = low-density lipoprotein cholesterol; HDL-C = high-density lipoprotein cholesterol; TG = triglyceride; WMD = weighted mean difference.

**Figure 3 foods-11-01982-f003:**
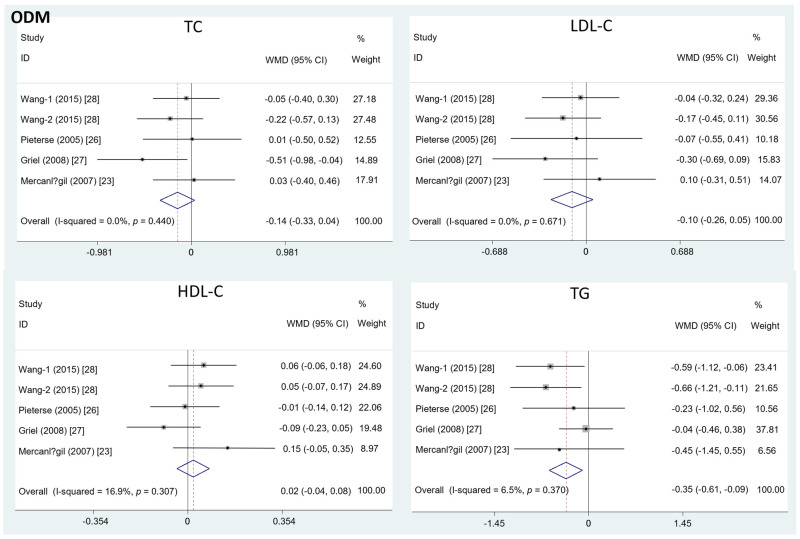
The effects of ODM on TC, LDL-C, HDL-C, and TG.TC = total cholesterol; LDL-C = low-density lipoprotein cholesterol; HDL-C = high-density lipoprotein cholesterol; TG = triglyceride; WMD = weighted mean difference.

**Figure 4 foods-11-01982-f004:**
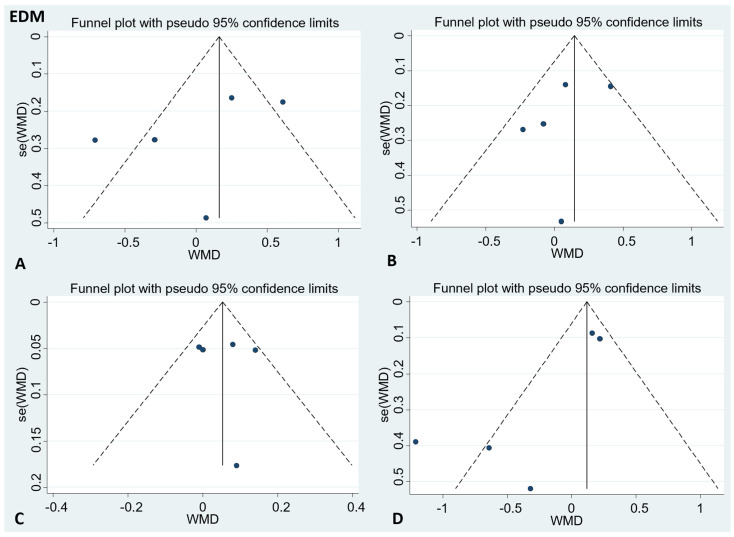
Funnel plots to evaluate publication bias, and the effect of EDM for (**A**) TC Egger’s test (*p* = 0.47), (**B**) LDL-C Egger’s test (*p* = 0.56), (**C**) HDL-C Egger’s test (*p* = 0.96), and (**D**) TG Egger’s test (*p* = 0.11). A = TC; B = LDL-C; C = HDL-C; D = TG; TC = total cholesterol; LDL-C = low-density lipoprotein cholesterol; HDL-C = high-density lipoprotein cholesterol; TG = triglyceride.

**Figure 5 foods-11-01982-f005:**
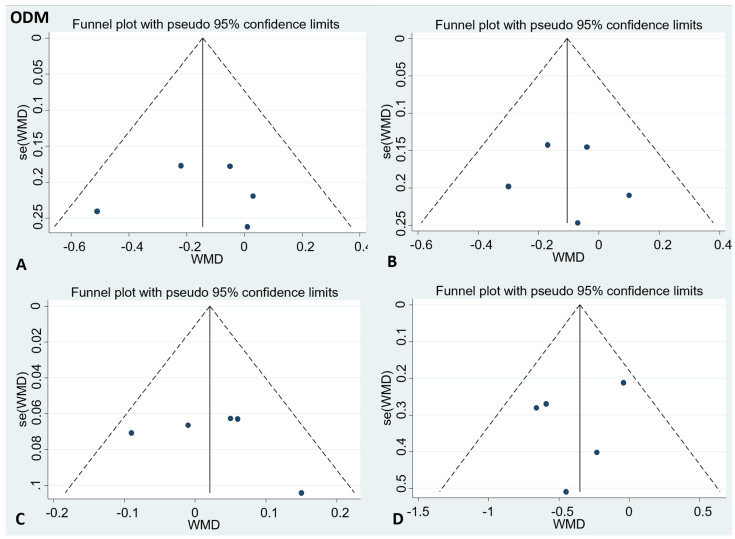
Funnel plots to evaluate publication bias, and the effect of ODM intake for (**A**) TC Egger’s test (*p* = 0.89), (**B**) LDL-C Egger’s test (*p* = 0.82), (**C**) HDL-C Egger’s test (*p* = 0.55), and (**D**) TG Egger’s test (*p* = 0.5). A = TC; B = LDL-C; C = HDL-C; D = TG; TC= total cholesterol; LDL-C = low-density lipoprotein cholesterol; HDL-C = high-density lipoprotein cholesterol; TG = triglyceride.

**Table 1 foods-11-01982-t001:** Baseline characteristics of participants and design characteristics of included articles.

Literature	Year	Study Region	Mean Age (Years)	Number of Participants (M/C)	MUFA Source	Amount of MUFA (M/C)	Study Design
Salar-1 [23]	2016	Iran (Asia)	51.42	24/23	Canola oil	6.9/6.5 ^a^	Parallel
Salar-2 [23]	2016	Iran (Asia)	51.42	25/23	Rice bran oil	7.63/6.5 ^a^	Parallel
Negele [24]	2015	Austria	11.1	12/9	Rapeseed oil	NA	Parallel
Welma-1 [26]	2021	Australia	32.66	20/22	Palm olein	25.65/9.42 ^b^	Parallel
Welma-2 [26]	2021	Australia	32.66	20/21	Palm olein	25.65/15.14 ^b^	Parallel
Wang-1 [28]	2015	America	45	42/43	Avocado	17/12 ^a^	Crossover
Wang-2 [28]	2015	America	45	43/43	High oleic acid oils	17/12 ^a^	Crossover
Pieterse [25]	2005	South Africa	40.8	28/27	Avocado	20/0 ^b^	Parallel
Griel [27]	2008	America	50.2	25/25	Nut	18/12 ^a^	Crossover
Mercanligil [22]	2007	Turkey (Asia)	48.0	15/15	Hazelnut	17–20/13–15 ^a^	Crossover

M: MUFA-rich diet, C: control diet; ^a^: the amount of daily MUFA intake (% of total energy); ^b^: the amount of daily MUFA intake (g).

**Table 2 foods-11-01982-t002:** Risk of bias assessment.

Study	Random Sequence Generation	Allocation Concealment	Blinding of Participants and Personnel	Blinding of Outcome Assessment	Incomplete Outcome Data	Selective Reporting	Other Sources of Bias
2016 Salar [24]	L	L	L	L	L	L	U
2015 Negele [25]	H	L	L	L	L	L	U
2021 Welma [27]	L	L	H	L	L	L	U
2015 Wang [28]	U	U	H	L	L	L	U
2005 Pieterse [26]	U	H	H	L	L	L	U
2008 Griel [27]	U	U	U	L	L	L	U
2007 Mercanlıgil [23]	U	U	H	L	L	L	U

L = low; H = high; U = unclear.

**Table 3 foods-11-01982-t003:** The results of subgroup analysis in included studies.

Subgroups	TC				LDL-C				HDL-C				TG			
	N	WMD (95%CI)	Heterogeneity		N	WMD (95%CI)	Heterogeneity		N	WMD (95%CI)	Heterogeneity		N	WMD (95%CI)	Heterogeneity	
			*p*	I^2^ (%)			*p*	I^2^ (%)			*p*	I^2^ (%)			*p*	I^2^ (%)
**MUFA source**	5	0.01 (−0.46, 0.48)	0.001	79.4	5	0.15 (−0.02, 0.32)	0.175	37.0	5	0.05 (0.01, 0.10)	0.200	33.2	5	−0.16 (−0.52, 0.20)	0.002	76.6
Canola oil	2	−0.2 (−0.67, 0.27)	0.520	0	2	0.17 (−0.64, 0.30)	0.638	0	2	0.01 (−0.09, 0.1)	0.624	0	2	−0.52 (−1.15, 0.11)	0.627	0
Palm olein	2	0.42 (−0.07, 0.78)	0.134	55.6	2	0.24 (0.04, 0.44)	0.101	62.8	2	0.11 (0.04, 0.17)	0.383	0	2	0.19 (0.06, 0.31)	0.654	0
Rice bran oil	1	−0.71 (−1.25, −0.17)	-	-	1	−0.08 (−0.57, 0.41)	-	-	1	−0.01 (−0.11, 0.09)	-	-	1	−1.21 (−1.97, −0.45)	-	-
**Participant condition**	5	0.01 (−0.46, 0.48)	0.001	79.4	5	0.15 (−0.02, 0.32)	0.175	37.0	5	0.05 (0.01, 0.10)	0.200	33.2	5	−0.16 (−0.52, 0.20)	0.002	76.6
Patients	3	−0.41 (−0.80, −0.02)	0.313	0	3	−0.13 (−0.47, 0.21)	0.864	0	3	0.00 (−0.07, 0.07)	0.860	0	3	−0.79 (−1.29, −0.29)	0.346	5.7
Healthy people	2	0.42 (−0.07, 0.78)	0.134	55.6	2	0.24 (0.04, 0.44)	0.101	62.8	2	0.11 (0.04, 0.17)	0.383	0	2	0.19 (0.06, 0.31)	0.654	0

TC: total cholesterol; LDL-C: low-density lipoprotein cholesterol; HDL-C: high-density lipoprotein cholesterol; TG: triglyceride; N: number of trials; WMD: weighted mean difference.

## Data Availability

Not applicable.

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
