# Peer review of "The Effect of MUFA-Rich Food on Lipid Profile: A Meta-Analysis of Randomized and Controlled-Feeding Trials"

_foods, 2022, doi:10.3390/foods11131982_

Round 1

Reviewer 1 Report

I reviewed the manuscript entitled, The effect of MUFA-rich food on lipid profile: A systematic review and meta-analysis of randomized, controlled-feeding trials. 

This review covered the role of MUFA-rich food on lipid profile with focused discussion.

Moreover, this topic filled an identified research gap addressing the MUFA enriched foods and lipid profile. Overall, the review is interesting, comprehensive and with relevant references, and established the need for future studies on lipid profile. Based on these observations, the review should be addressed below suggestions

Title: can be revised as “The effect of MUFA-rich food on lipid profile: A systematic review, meta-analysis of randomized, and controlled-feeding trials”

Why did authors not cover the systematic reviews or research papers? Also, provide the exact date and year of information retrieval. Every day there are many papers that have been added to databases.

Need of conducting meta-analysis should be discussed

The abstract is prepared as a structured abstract. This is not the abstract format in foods. Please revise it.

Please revise the references according to journal format

Reviewer 2 Report

The present study includes a meta-analysis of the effect of MUFAS on serum lipids.

The overall analysis is excellent since it performs a filter that allows finding only relevant studies associated with the central theme. Thanks to this type of study we can find true results within the scientific literature and at the same time, it shows us the importance of homogenizing the studies, in this case, the consumption of MUFAS, and thus be able to understand its true effect on health.

On the other hand, although the analysis and results obtained are good, the authors should correct the presentation of the manuscript.

Here are some observations:

Correct eu-rope line 42

Correct "toto" line 60

Correct ac-id line 64

Please like the previous observation, correct all words that have unnecessary hyphens throughout the manuscript.

The meaning of the letters in all tables should be explained.

Improve the resolution of figures and also explain the meaning of letters.

In table 3, is palm olein?

Correct unnecessary spaces between words

Reviewer 3 Report

Comments on foods-1758840.

There are many studies that have shown associations between MUFA consumption and the lipid profile. It remains unclear how much of that is cause or effect. The authors have tried to look into this trough a meta-analysis of the different trials that have been published. I think that this is relevant and useful approach but I am not convinced about the results.

The inclusion and exclusion criteria are unclear and I do not understand why the authors have included the data on canola oil for the Salar paper but not those for rice bran oil (which contains a high amount of MUFA (48%)). The authors need to make clear what difference between the control diet and test diet relates to relevant increase in MUFA intake.

Secondly, I strongly disagree with combining the results for EDM and ODM in the conclusion. These are very separate concepts. As the effect of Avocados and hazelnuts on the lipid profile can be caused by many different things then just MUFA content and the discrepancy between EDM and ODM confirms this. I would therefore suggest that the discussion will only be focussed on the EDM and the ODM results can be used as comparison and additional information but that this does not provide any evidence on the effect of MUFA on the lipid profile.

Furthermore it is essential that there is more emphasis on the power of the studies and the number in the groups and their age range.  Also the M&M needs to give more detail on how different numbers have been calculates (such as weight %, these vary hugely between the different lipids while the I2 only is above 50% for a few.

Tables 2 and 3 miss information of the acronyms used  (L H U etc) or WMD.

I would suggest to exclude Welma-1 from the HDL meta analysis as it has been shown by different studies that coconut fat leads to a decrease in HDL. The suggested effect of MUFA in this study is misleading. 
